# Laser Microperforation Assisted Drug-Elution from Biodegradable Films

**DOI:** 10.3390/pharmaceutics14102144

**Published:** 2022-10-09

**Authors:** Arkady S. Abdurashitov, Pavel I. Proshin, Olga A. Sindeeva, Gleb B. Sukhorukov

**Affiliations:** 1V. Zelmann Center for Neurobiology and Brain Rehabilitation, Skoltech, 121205 Moscow, Russia; 2School of Engineering and Materials Science, Queen Mary University of London, London E1 4NS, UK; 3Siberian State Medical University, Moskovskiy Trakt, 2, 634050 Tomsk, Russia

**Keywords:** drug-eluting coatings, biopolymers, zero-order elution, controlled release, laser microperforation

## Abstract

In a modern high-tech medicine, drug-eluting polymer coatings are actively used to solve a wide range of problems, including the prevention of post-surgery infection, inflammatory, restenosis, thrombosis and many other implant-associated complications. For major assumptions, the drug elution mechanism is considered mainly to be driven by the degradation of the polymer matrix. This process is very environmentally dependent, unpredictable and often leads to a non-linear drug release kinetic. In the present work, we demonstrate how the laser microperforation of cargo-loaded biodegradable films could be used as a tool to achieve zero-order release kinetics with different elution rates. The effects of the laser-induced hole’s diameter (10, 18, 22, 24 µm) and their density (0, 1, 2, 4 per sample) on release kinetic are studied. The linear dynamics of elution was measured for all perforation densities. Release rates were estimated to be 0.018 ± 0.01 µg/day, 0.211 ± 0.08 µg/day, 0.681 ± 0.1 µg/day and 1.19 ± 0.12 µg/day for groups with 0, 1, 2, 4 microperforations, respectively. The role of biodegradation of the polymer matrix is reduced only to the decomposition of the film over time with no major influence on elution rates.

## 1. Introduction

Drug-eluting coatings are an attractive tool for the modification of various implantable medical devices for long-term drug delivery and local therapy [1,2,3,4]. However, the characteristic drug-release time is affected by many factors, including polymer type [5], pH of the environment [6,7], chemical properties of the drug, etc. [8]. In addition, most biopolymers have relatively large water vapor permeability in their bare state [9,10]. In combination with relatively low coating thickness (tens of microns or less), it means that dry hydrophilic cargo after the implantation in the living body will be in the form of a highly concentrated solution or at least stored at high relative humidity. Most medicines are unstable in such conditions and could only be stored for a few days or a little over a week, especially antibiotics [11]. Ideally, the cargo should be eluted during the safe period, when no chemical decomposition of the medicine has occurred. This could be accomplished by tuning the drug permeability properties of the polymer film by either accelerating its biodegradation in time [12] or chemically inducing porosity, thanks to which the drug could be eluted [13]. These processes of film modification are time consuming and heavily environmentally dependent. When the external conditions are changed, the tuning process has to be done again to achieve the desired result. For example, if the pH value of the surroundings has been changed, the polymer biodegradation could be substantially slowed and no pores could be formed, which will lead to the violation of the desired drug-elution kinetic.

Here, we described a simple modification of drug-eluting films, which were allowed to break the link between the biodegradation of the polymer and the release kinetic by introducing laser microperforation of the coating. By adjusting the diameter of the holes and their density, a wide range of elution kinetics can be accomplished independently from the biodegradation of the polymer. For the base polymer, we chose a poly(lactic-co-glycolic acid) (PLGA), with the lactic/glycolic ratio being 50:50. It is a biocompatible and biodegradable polymer widely used in the coating of medical devices and in medicine in general, including particles and implants [2,14,15]. PLGA exhibits a strong tendency to hydrolysis and will degrade gradually in a course of a few months within the aquatic medium [16]. PLGA is a water vapor permeable polymer [17]; however, to the author’s knowledge, it is impenetrable for most medically active components, especially biomacromolecules. These facts give a possibility to avoid the consideration of the biodegradation and membrane impacts on a drug-elution kinetic in the case of PLGA-based coatings in normal environmental conditions. Drug elution will be mostly governed by the diffusion through the laser-made holes of a given diameter from the medium with a high drug concentration (polymer coating) to the drug-free environment (soft tissues of the body or model liquid), which makes it predictable and controllable.

## 2. Materials and Methods

### 2.1. Manufacturing Routine of a Drug-Eluting Films

PLGA (PURASORB^®^ PDLG 5010) was dissolved in a 80:20 mixture of ethyl acetate (Acros Organics, Geel, Belgium) and acetone (Acros Organics) to form a 10 wt% polymer solution. The dissolving process was performed in a heated water bath (40 °C) with moderate stirring for 2 h. After that, the solution was cooled down to room temperature.

The base film was obtained by applying a polymer solution to a polypropylene (PP) substrate (40 µm thick) via the Baker film applicator machine (blade gap 100 µm; application speed 20 mm/s). The base film was dried for 15 min at 50 °C.

Tetramethylthionine chloride (also known as methylene blue or MB) (Sigma-Aldrich, St. Louis, MO, USA) was chosen as a drug-model substance. It has a modest water solubility of 43.6 g/L, and its content in the solution could be quantified by the spectroscopic analysis.

A total of 20 mg of MB were added into a 1 mL 9 wt% solution of poly(vinyl alcohol) (PVA) (Sigma Aldrich, St. Louis, MO, USA) and firmly mixed. Air bubbles were evacuated using vacuum pump. MB containing PVA gel was loaded into the 1 mL syringe and inserted into the automated dispenser device. The gel was deposited onto the base film in a form of 1 × 1 cm squares filled with snake-like pattern using a G23 needle with a volumetric flow rate of 0.85 mL/h and left to dry (30 min at 50 °C).

To seal the dry gel, a cover layer was formed using the above mentioned 10 wt% PLGA solution via the Baker film applicator (blade gap 400 µm; application speed 20 mm/s) and left for 15 min in a room environment for drying.

The completed coating consisting of a base film and a dried gel layer, and a cover film was left on a PP substrate and placed into the vacuum oven (100 mBar, 40 °C) for 2 h to complete the drying process.

A graphical description of the film’s manufacturing process is shown in Figure 1a.

### 2.2. Laser Microperforation Process

Cobolt Tor^TM^ XS 532 nm pulsed laser (50 µJ; 1.9 ns) was used to perform a laser microperforation of the biopolymer films. Laser irradiation was focused by the objective lens (8 × 0.2) onto the film’s surface. Three consecutive laser pulses were utilized to perform one microperforation (pulse repetition rate is 1 kHz). The films were positioned under the laser beam using a precision motorized XY stage (STANDA LTD, Vilnius, Lithuania) with an accuracy of ±1 µm.

### 2.3. Assessment of the Drug Elution Kinetics

The spectroscopic study was conducted using TECAN Infinite^®^ 200 Pro M Nano Plus plate reader to analyze the elution kinetics. MB loaded samples were placed into 1 mL of saline (0.9% NaCl in deionized water) and kept in hermetically sealed eppendorf containers under heat and shaking (37 °C; 300 RPM) for 24 h to mimic the body fluid dynamic. After 24 h, a small portion (150 µL) was taken out from the eppendorf to determine the dye content. The samples were transferred into the same volume of fresh saline every 24 h. This process was repeated for one week to asses the middle-term elution data.

### 2.4. Effect of Hole Diameter on the Release Rate

To assess the effect of the hole diameter on the elution kinetics, a set of holes with known diameters (24, 22 and 18 µm) were produced in the MB-loaded polymer film by the laser ablation. To quantify the release of a substance through microperforations, the sample was placed in a custom made flow cell. Details about the flow cell are listed in the supporting information (Section A.1). A peristaltic pump was used to pump the saline (0.9% NaCl in H_2_O, 37 °C) through the cell for 24 h with a rate of 10 mL/min. The time-lapse of MB elution was recorded using a DCC1545M CMOS camera (THORLABS, Inc.: Newton, NJ, USA) and M112FM25 imaging lens (TAMRON). One frame of the sample was taken every 10 s. A white light emitting diode (LED) was used as the illumination source. A long-pass filter with a cut-on wavelength of 630 nm was utilized to achieve a good image contrast while observing MB elution. Image processing was performed using a custom made Python code (Python 3.7.0; Numpy 1.19.1; Opencv-python 4.3.0.36).

### 2.5. Sample Characterization

Morphological studies of the surface of samples were performed using a TESCAN Vega 3 scanning electron microscope (SEM) and Olympus CX33 transmission optical microscope.

## 3. Results

### 3.1. Film’s Dimensions and Payload

A set of MB loaded films were manufactured from PLGA polymer according to Section 2.1. Figure 1b indicates the typical optical image of the sample. According to the SEM analysis of the film’s cryoslices, the average thicknesses of the base films, cargo layer, and the cover film were 3 ± 1 µm, 5 ± 1 µm, 14 ± 1 µm, respectively (Figure 1c). Total load of MB was measured to be 35 ± 9 µg/cm${}^{2}$.

### 3.2. Laser Microperforation of a MB Loaded Biodegradable Film

Experimental data demonstrated a linear relationship between the laser radiation energy and the diameter of the resulting hole within the energy levels from 15 to 50 µJ (Figure 2). For laser energies of 15, 35, and 50 µJ measured diameters of the holes are 18 ± 2.3 µm, 22 ± 2 µm, and 24 ± 2.5 µm, respectively. Dimensions were taken from the optical image using an ImageJ software. Although the wavelength of the used laser source (532 nm) lies outside the absorption range of the PLGA, it is well absorbed by the MB dye. The plasma ball, which is formed as a result of the absorption of high-intensity laser radiation by the dye, damages the top layer of the coating, producing a hole. There is a relatively small difference in mean diameters (18, 22, and 24 µm) of laser made holes, no more than 25% deviation. However, the affected area scales significantly, as it grows as a diameter squared, so the cross section of the hole scales up to ∼1.7 folds.

### 3.3. Effect of Hole Diameter on Elution Rate

Figure 3 highlights results computed by the analysis of image sequence obtained in a flow cell experiment. There is a clear difference between the width of the elution zones (>90% eluted cargo) for laser-made holes with different diameters, as indicated by the Figure 3c). Although on average the sample released 50% of its total payload, zones of certain width are formed near the laser made microperforations of different diameters. The widths of such zones are 860 ± 50 µm, 625 ± 45 µm, and 300 ± 40 µm for hole diameters of 24 µm, 22 µm, and 18 µm, respectively (Figure 3e). To compute the relative amount of eluted MB, the initial average intensity of the MB stripe was taken as a 0% elution ($I_{E0}$) and the average intensity of the MB-free surrounding was taken as 100% elution ($I_{E100}$). The intensity of each pixel ($I_{x,y}$) in the image of the MB-loaded film taken after 24 h in the flow cell were converted to the relative amount of eluted MB (EC) by the Equation (Equation 1): (1)$$EC\left[\%\right]=\frac{I_{x,y}-I_{E0}}{I_{E100}-I_{E0}}*100.$$

The relative elution rate (ER) of the MB was obtained by performing an analysis of time-dependant intensity of each pixel within the image ($I_{x,y,t}$). At first, each intensity value in the time-series was converted to relative elution according to the Equation (Equation 1).Then, the rate of relative elution change was computed by calculating the discrete difference between the adjacent time-points by the Equation (Equation 2): (2)$$ER\left[\%/min\right]\left(t\right)=\frac{I_{x,y,t_{i}}-I_{x,y,t_{i+1}}}{\Delta t}.$$

Maximum relative elution rate was picked for each image pixel to construct a 2D map shown in Figure 3d. A high elution rate (∼0.6%/min) was observed near the laser made holes with a gradual reduction to ∼0.1%/min as the distance from the hole increases. The width of the zone at which the relative elution rate falls to the average value correlates with the width of the zone of complete MB release after 24 h in the flow cell.

### 3.4. Quantification of the MB Elution Depending on the Number of Holes

To quantify the amount of eluted MB, a spectrophotometric calibration was performed. At low concentrations (<10 µg/mL) MB exhibits a linear dependency between the absorption value at 665 nm and its concentration (Figure A2).

A set of samples were prepared according to Section 2.1 to study the effect of the number of holes on the elution rate. A hole with the diameter of 10 µm was chosen according to a flow cell experiment, as holes with mean diameters of <18 µm are feasible for middle-term or long-term drug release applications. The samples were divided in four groups:no holes (n = 6);1 hole (n = 6);2 holes (n = 6);4 holes (n = 6).

The quantitative data of MB elution depending on the number of holes are shown in Figure 4a. Locations of laser-made holes for different groups of samples are indicated in Figure 4b. All groups demonstrated linear (also known as zero-order) elution kinetics. The first Fick’s law could be used to mathematically describe this process, as follows:(3)$$\frac{dM}{dt}=\frac{DAS}{h},$$
where dM/dt is the mass change rate, *D* is the diffusion coefficient for the pair of solute and solution, *S* is the total area of microperforations, and *h* is the thickness of the cargo layer. Elution rates were measured to be 0.018 ± 0.01 µg/day, 0.211 ± 0.08 µg/day, 0.681 ± 0.1 µg/day, and 1.19 ± 0.12 µg/day for groups with 0, 1, 2, 4 microperforations, respectively. The main mechanism of linear elution could be described as follows: near the laser-made hole, a small volume from which the drug could diffuse to the environment is created. On the other hand, there is a constant substance supply to this volume from the rest of the sample. This supply is possible due to the swelling of dye containing PVA gel with the water molecules, permeated through the cover film. These factors are balanced in such a way that the zero-order elution is achieved. The linear dependency was observed (Figure 5) between the number of holes and elution rate with the slope of ∼0.3 µg/day/hole. Given the average total load of MB per sample (35 ± 9 µg/cm${}^{2}$), we can consider a theoretical characteristic release time from a month-long to years depending on the number of perforations.

## 4. Discussion

In this work, we describe a simple method of the modification of drug-eluting films to achieve zero-order elution kinetics with different rates. The solubility of used dye (43 g/L in water) is on-par with real medicine, namely antibiotics or other highly bioavailable drugs. This fact gives us a reason to expect the same elution behavior for bioactive cargos, as long as there is no chemical interaction between the base polymer and the cargo, which could accelerate the biodegradation. For the highly water soluble substances (>200 mg/mL), faster release is expected, while maintaining linearity according to Equation (Equation 3). Faster rates could be compensated by the reduction of the hole’s diameter. Although the years-long elution requires the usage of stable base polymers such as polyimide, polyethylene or polytetrafluoroethylene to achieve a stable performance, months-long elution is practically feasible and could be accomplished by the wide range of biopolymers such as PLGA, PLA, PCL, and many others. It is expected that on the large time scale (several months), the elution kinetics will cease to be linear. This is due to the fact that as the elution progresses, the substance’s molecules have to travel longer distances within the cargo layer until they reach a hole and diffuse to the environment. This non-linearity in the long-term elution could be addressed by reducing the overall cargo quantity and increasing the density of the perforations. This will allow for the prolongation of the zero-order elution regime of the described coatings, as travel distances for cargo molecules will be limited by the maximum distance between adjacent laser-made holes.

A quantitative and qualitative assessment of the release rate of the substance was carried out under the high flow rate (10 mL/min) and in almost static conditions (1 mL/day). High flow environment expectantly accelerates the elution rate. Up to 50% of the initial cargo was released in the first 24 h, compared to a few percent over the course of a week in the case of static conditions. In addition to this, the flow cell experiment showed the formation of complete release zones inside the cargo layer (Figure 3e), which were not observed in the case of substance elution under almost static conditions (1 mL/day). This effect can be explained by the fact that fast fluid flows rapidly wash out the substance from the cargo layer and the speed of MB diffusion within the cargo layer is not enough to compensate for this.

Drug delivery microsystems with zero-order elution kinetics are very attractive and demanded in modern pharmacy [18] because of their ability to maintain a drug level between the minimum effective concentration (MEC) and minimum toxic concentration (MTC) for a desired time.

The main advantage of our system is the payload capabilities. Depending on the drug’s solubility, high mass of cargo (up to 1 mg/cm${}^{2}$) could be loaded in proposed coatings with little effect on the elution kinetics. In contrast, matrix systems such as composites are limited in their payload capabilities by 20–30% of the polymer mass. A higher cargo quantity will likely lead to volumetric erosion of the polymer matrix through the hydrophilic cargo “bridges”. This erosion would result in an accelerated release profile with an initial burst of cargo concentration in the surrounding environment. Such behavior of drug delivery systems is undesirable.

Authors are familiar with other drug delivery systems, which achieve zero-order elution via the help of drug administration through a hole of a given area such as elementary osmotic pumps [19,20], perforated microtubes [21] or microchips [22,23]. The manufacturing process of such devices is complex and multistage. Along with that, in the case of elementary osmotic pumps, their performances are highly affected by the environmental conditions. While perforated microtubes are capable of providing a long-term zero-order release, elementary osmotic pumps fell short after several hours (<12 h) [24].

A distinctive feature of our approach is the combination of simple manufacturing routine and the possibility of adjusting the release over a wide range of rates. Polymer films for the base and cover layers could be manufactured using a high throughput and well-developed method such as slot-die coating. Cargo layer could be deposited in any pattern to achieve a desired drug dosage or to distribute a fixed amount of drug over a given area. Laser microperforation can be applied using high speed beam scanning systems. Proposed modification of the biopolymer films can be used as free-standing drug administration devices (wound dressing) or fixed on medical devices for target drug delivery.

Most modern antibiotics suggest a duration of treatment within a week; in severe cases, treatment can be extended up to two weeks. This fact allows us to consider the proposed coatings as a promising tool for local antibacterial treatment with controlled constant dosage.

Various external stimuli such as therapeutic ultrasound or an external electric field are encouraged for future studies for an on-demand increase in the amount of substance released from the described coatings to potentially relieve the crisis conditions.

## 5. Conclusions

Laser microperforation have been successfully applied to control the substance elution rate from biopolymer coatings. Zero-order release kinetics were observed for each group of samples with 0, 1, 2, and 4 microperforations, respectively, for one week period. A linear dependency between the number of holes and elution rate was demonstrated. A simple, scalable, repeatable, and cost-effective method of production and modification of drug-eluting films for controlled drug administration has been demonstrated, and foreseen research and applications are exposed.

## Figures and Tables

**Figure 1 pharmaceutics-14-02144-f001:**
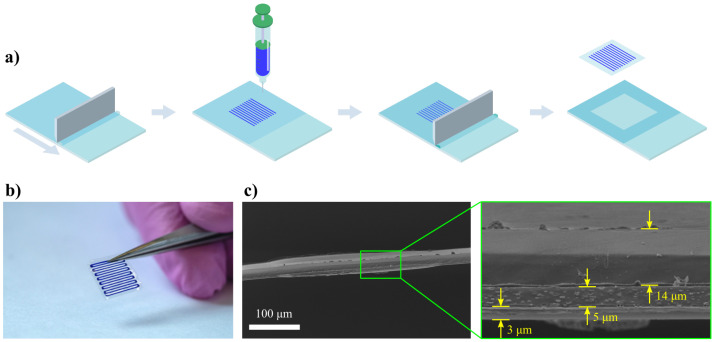
(**a**) Manufacturing routine for the production of drug-eluting films, including base film application, automatic cargo dispensing, and sealing. (**b**) Typical appearance of a sample used in this research. (**c**) SEM image of the internal structure of the film.

**Figure 2 pharmaceutics-14-02144-f002:**
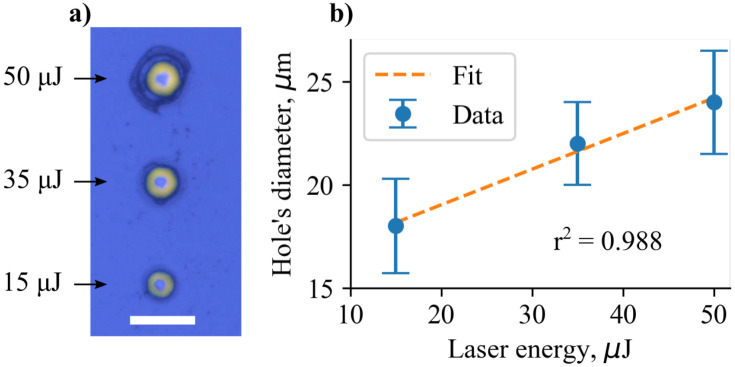
(**a**) Optical image of microperforations obtained at different laser energies. Scale is 50 µm. (**b**) The dependence between laser energy and produced hole diameter for the PLGA-based MB-loaded films. Ten holes were produced at each laser energy and their diameters were averaged to obtain a mean value.

**Figure 3 pharmaceutics-14-02144-f003:**
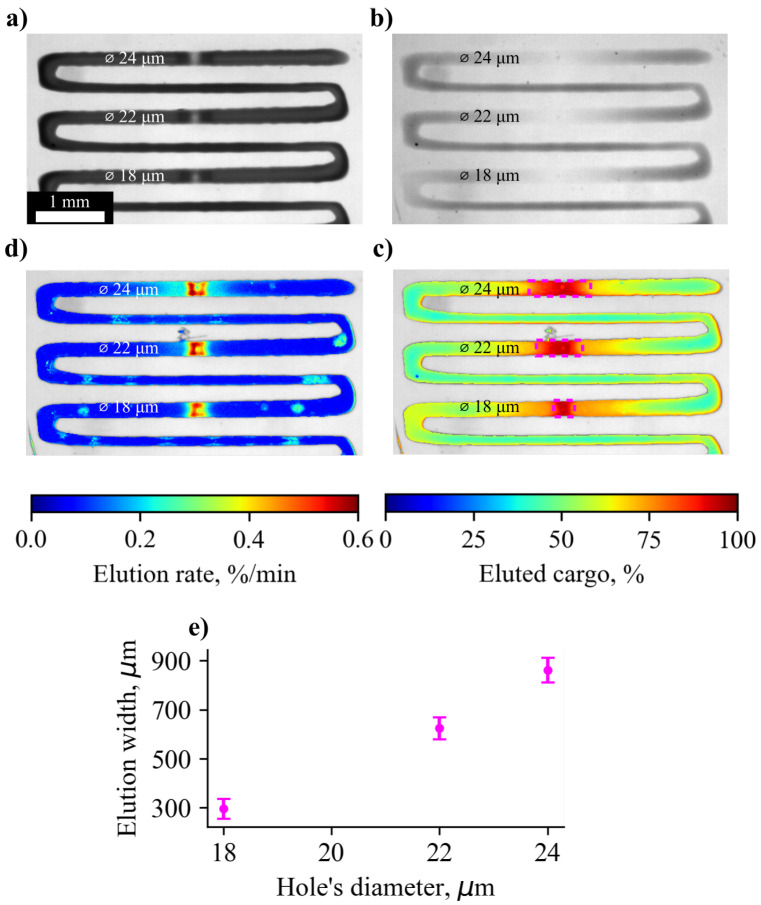
(**a**) Initial state of an MB loaded film placed in a flow cell. (**b**) Visual appearance of an MB loaded film after 24 h in a high flow environment. (**c**) Pseudo-colored map of the amount of eluted cargo from an MB-loaded film with microperforations of different diameters after 24 h in a high flow environment. (**d**) Pseudo-colored map of maximal observable elution rate for the holes with different diameters and their neighborhood. (**e**) The dependence between the hole’s diameter and the width of the elution zone with high amount of released substance (>90%).

**Figure 4 pharmaceutics-14-02144-f004:**
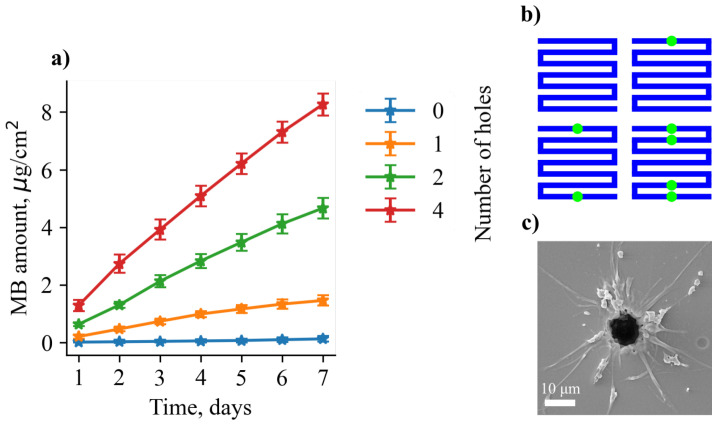
(**a**) One week MB elution data depending on the number of laser made holes. (**b**) Locations in which holes are made for different groups of samples—0, 1, 2, 4 holes, respectively. Locations are indicated as green circles. (**c**) SEM image of the laser-induced hole. Mean hole diameter is 10 µm.

**Figure 5 pharmaceutics-14-02144-f005:**
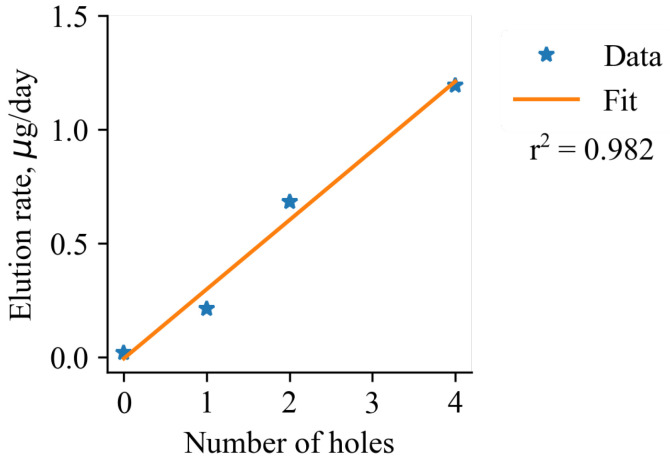
The linear dependence between the elution rate and the number of holes (∅18 µm).

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
