# Peer review of "Laser Microperforation Assisted Drug-Elution from Biodegradable Films"

_pharmaceutics, 2022, doi:10.3390/pharmaceutics14102144_

Round 1

Reviewer 1 Report

The authors developed a reservoir kind of drug delivery system, where the release can be controlled based on the number of holes inserted in the film via laser. As discussed by the authors themselves, the concept is not novel but the way it is fabricated simplifies the manufacturing process. However, I wonder, how this system is better than the matrix system. PLGA used in this study was shown to be degraded within two months and the degradation ratio can be easily tuned based on the lactide and glycolide ratio.  

Further, how does the presence of holes affect stability as it becomes easier e.g., oxygen to permeate inside?   The author claimed that these devices are easier to make but so are the matrix systems. 

Author Response

 Rev1:

Comments and Suggestions for Authors

The authors developed a reservoir kind of drug delivery system, where the release can be controlled based on the number of holes inserted in the film via laser. As discussed by the authors themselves, the concept is not novel but the way it is fabricated simplifies the manufacturing process. However, I wonder, how this system is better than the matrix system. PLGA used in this study was shown to be degraded within two months and the degradation ratio can be easily tuned based on the lactide and glycolide ratio.  

Further, how does the presence of holes affect stability as it becomes easier e.g., oxygen to permeate inside?   The author claimed that these devices are easier to make but so are the matrix systems. 

Answer:

The main advantage of our system is payload capabilities. Depending on the drug’s solubility, a high mass of cargo (up to 1 mg/cm2) could be loaded in proposed coatings with little effect on the elution kinetics. In contrast, matrix systems are limited in their payload capabilities by the 20-30% of the polymer mass. A higher cargo quantity will likely lead to a volumetric erosion of the polymer matrix through the hydrophilic cargo “bridges”. This erosion would result in an accelerated release profile with an initial burst of cargo concentration in the surrounding environment. Such behavior of drug delivery systems is undesirable.

PLGA is both oxygen and water vapors permeable polymer. If the payload is soluble in water in volume, it occupies in the polymer film, cargo will be inevitably stored in the form of concentrated solution with. In general, this is considered as “unsafe” storage conditions. Through the perforation mesh of a given density and hole’s diameter, we ensure a steady drug elution within the safe period prescribed by the drug manufacturer.

We added corresponding text in the manuscript in the discussion section on page 7 and marked it in cyan.

Reviewer 2 Report

This manuscript presents a novel idea of utilizing laser microporation to control the elution of polymers and thus leading to a zero kinetics release.

The manuscript can be apt for publication after perforimg the following:

1- Please revise the English language throughout the text as there are many mistakes such as "an linear", "an non-linear" and so on. Also the use ofcommas and punctuatuions should be thoroughly revised.

2- My main concern about Figure 2 is that it reveals the absence of significant differences between the points so it seems that the laser's different energies do not affect the holes' diameter. Please comment.

Author Response

Rev2:

Comments and Suggestions for Authors

This manuscript presents a novel idea of utilizing laser microporation to control the elution of polymers and thus leading to a zero kinetics release.

The manuscript can be apt for publication after perforimg the following:

  • Please revise the English language throughout the text as there are many mistakes such as "an linear", "an non-linear" and so on. Also the use ofcommas and punctuatuions should be thoroughly revised.

Answer:

We checked the manuscript for spelling and punctuation errors and corrected them

2- My main concern about Figure 2 is that it reveals the absence of significant differences between the points so it seems that the laser's different energies do not affect the holes' diameter. Please comment.

Answer:

In figure 2, there is the optical image of typical laser made holes producing with different pulse energies. Indeed, there is a relatively small difference in mean diameters (18, 22 and 24 µm), no more than 25% deviation. However, the affected area scales significantly, as it grows as diameter squared, so the cross section of the hole scales up to 1.7 folds leading to a different elution rates, observed in a flow cell experiment.

We added corresponding text in the manuscript in the 3.2 section on page 4 and marked it in cyan.